# Fluid mechanics of the zebrafish embryonic heart trabeculation

**Adriana Gaia Cairelli**[1], **Renee Wei-Yan Chow**[2], **Julien Vermot**[1,2], **Choon Hwai Yap**[1] *

**1** Dept of Bioengineering, Imperial College London, London, United Kingdom, **2** Institut de Génétique et de Biologie Moléculaire et Cellulaire (IGBMC), Strasbourg, France

* c.yap@imperial.ac.uk

**Data Availability Statement:** The images and ANSYS simulation files including user defined functions are provided at https://doi.org/10.5281/zenodo.6572930. The source code for the image

## Abstract

Embryonic heart development is a mechanosensitive process, where specific fluid forces are needed for the correct development, and abnormal mechanical stimuli can lead to malformations. It is thus important to understand the nature of embryonic heart fluid forces. However, the fluid dynamical behaviour close to the embryonic endocardial surface is very sensitive to the geometry and motion dynamics of fine-scale cardiac trabecular surface structures. Here, we conducted image-based computational fluid dynamics (CFD) simulations to quantify the fluid mechanics associated with the zebrafish embryonic heart trabeculae. To capture trabecular geometric and motion details, we used a fish line that expresses fluorescence at the endocardial cell membrane, and high resolution 3D confocal microscopy. Our endocardial wall shear stress (WSS) results were found to exceed those reported in existing literature, which were estimated using myocardial rather than endocardial boundaries. By conducting simulations of single intra-trabecular spaces under varied scenarios, where the translational or deformational motions (caused by contraction) were removed, we found that a squeeze flow effect was responsible for most of the WSS magnitude in the intra-trabecular spaces, rather than the shear interaction with the flow in the main ventricular chamber. We found that trabecular structures were responsible for the high spatial variability of the magnitude and oscillatory nature of WSS, and for reducing the endocardial deformational burden. We further found cells attached to the endocardium within the intra-trabecular spaces, which were likely embryonic hemogenic cells, whose presence increased endocardial WSS. Overall, our results suggested that a complex multi-component consideration of both anatomic features and motion dynamics were needed to quantify the trabeculated embryonic heart fluid mechanics.

## Author summary

In the embryonic heart, the mechanical forces that blood fluid imposes on the cardiac tissues are known to be important biological stimuli that affect the proper heart development. We thus perform careful quantification of these forces, using the zebrafish embryo as a model. To do this, we perform high resolution imaging of zebrafish embryonic hearts and image-based flow simulations. We find that the use of a particular fish line that

cardiac motion tracking is available at https://github.com/WeiXuanChan/motionSegmentation.

**Funding:** This study is supported by the Imperial College PhD Scholarship for supporting AGC, EU Horizon 2020 Research and Innovation Program GAN0682938 (JV), Agence Nationale de la Recherche grants ANR-SNF 310030E-164245 (JV), ANR-10-LABX-0030-INRT (JV), and ANR-10-IDEX-0002-02 (JV), the Foundation Lefoullon Delalande 2019 for supporting RC, and Imperial College startup funding (CHY). The funders had no role in study design, data collection and analysis, decision to publish, or preparation of the manuscript.

**Competing interests:** The authors have declared that no competing interests exist.

expresses fluorescence at the exact boundary between heart tissue and blood, that is the endocardial cell membrane boundary, is important to give high quality results. The heart's inner surface has uneven trabeculation structures. We find that they cause fluid forces to have spatial variability and an oscillatory nature. We also find that there is a squeezing motion of cardiac tissues on the trabeculation fluid spaces, which is the main mechanism that generated fluid forces. Fluid forces are also affected by a number of cardiac cells that were developing into blood cells, lodged in the trabeculation fluid spaces. Our investigations provide an understanding of the complexity of the fluid forces on the inner surface of the embryonic heart, and our quantifications will be useful to future studies on the biology elicited by these fluid forces.

## Introduction

During the development of the ventricle, the myocardium differentiates into two layers, an outer compact zone and an inner trabeculated zone, both of which are essential for normal cardiac contractile function. The trabeculae are myocardial cells covered by an endocardial layer, and they form a network of complex and undulating endocardial surface structures that can have a significant influence on flow dynamics near the ventricular walls.

Cardiac trabeculation is likely to be important for proper cardiac function. The trabeculae are thought to enhance nutrient transport to the heart prior to the formation of the coronary vessels by increasing the surface area available for biotransport [1], and seem to serve as a fast conduction system in cardiac electrophysiology [2]. Defects in trabeculations or inhibition of genes related to their formation led to embryonic deaths [3–5], further suggesting that they are important for life.

Past studies have shown that ventricular chamber maturation and trabeculae formation are mechanosensitive processes [6–8], and disrupted fluid mechanical stimuli lead to abnormal cardiac morphogenesis and malformations [9]. For example, decreased stresses that ventricular blood flow imposes on the endocardium in the *weak atrium mutants* and *gata1* morphants impedes the formation of trabeculae [10,11]. Furthermore, the elimination of cardiac contractility and blood flow via *tnnt2a* morpholino and *2,3-butanedione monoxime* (BDM) downregulates the Neuregulin 2a (Nrg2a) [12], which is essential for trabeculation, and leads to disruption of trabeculae formation. Prolonged disruption of contractility via BDM also leads to abnormal cardiac looping, underdevelopment of the heart, and reduced cardiac function [13]. Since flow biomechanical forces are important for the cardiac development, it is essential to accurately characterize fluid forces in the heart to enable further advancements.

For this reason, many investigators have performed quantifications of fluid wall shear stresses (WSS) on the zebrafish embryonic heart endocardium. However, accurate estimation of the endocardial flow WSS requires considerations for the complex 3D geometry and motion dynamics of the trabeculated endocardial boundary. Jamison *et al.* performed 2D particle image velocimetry measurements using blood cells as tracer particles, but did not use an imaging approach that could provide detailed trabeculation structures [14]. Battista, Miller and coworkers conducted computational fluid dynamics (CFD) simulations, but performed 2D simulations and used idealized instead of image-based geometries [7,15,16]. Foo *et al.* performed 4D CFD simulations based on 4D microscopy images, but used smoothed-out endocardial boundaries with no consideration of trabecular geometry [17]. Lee *et al.* and Vedula *et al.* conducted moving boundary CFD simulations based on 4D images from light-sheet fluorescence microscopy, and included considerations for trabeculation structures [9,10,18]. They

concluded that WSS within the intra-trabecular space had higher oscillatory shear index (OSI), and were of lower magnitude than WSS at the ridges of the trabeculation structures. However, they reconstructed the endocardial boundary geometry from fluorescent images of the myo-cardium instead of endocardium.

In the current study, we sought to build upon these existing works and achieve an improved quantification of the zebrafish embryonic endocardial flow stresses. We performed 4D mov-ing-boundary CFD simulations for this quantification, based on high-resolution microscopy imaging of a transgenic fish line with endothelial membrane fluorescent markers, which allows clearer imaging of fine details of trabecular surface geometries and motion dynamics. We also conducted simulations to show that the motion dynamics of the endocardial boundary is important for this quantification.

## Results

### Imaging and motion tracking results

Images of the 3 days post fertilization (dpf) embryonic heart of our fish line with endocardial cell membrane fluorescent markers demonstrated significant surface unevenness due to trabe-culation. Manual quantification from the images revealed that intra-trabecular spaces had lon-gitudinal, radial and circumferential dimensions of 12.5±3.1 μm, 8.7±0.8 μm, 21.7±10.7 μm (*n = 16*) respectively (measurements were taken at end systole). More surface features seemed to be observed from our images than in previous studies where myocardial rather than endo-thelial fluorescent markers were used [10,18]. This could be due to the use of confocal imaging as opposed to conventional fluorescent microscopy, and the use of a zebrafish line with fluo-rescence only at the endocardial cell membrane that could have enabled clearer identification of endocardial-blood boundaries. Further, we used a validated motion tracking image process-ing algorithm [19] on the images, and Fig 1 and S1 Movie could show that high fidelity track-ing of endocardial boundary motions was achieved.

### Characterization of WSS in the whole ventricle

We performed moving-boundary CFD flow simulations, using the 4D images (3D over time) of embryonic hearts. WSS results of the whole ventricular simulations are shown in Fig 2A, Fig C in S1 Text and S2 Movie, for the scenario where fluid was assumed to have viscosity of plasma (1.5 cP). Compared to the simulation scenario that assumes the viscosity of whole blood (7.35 cP), as shown in Fig G in S1 Text and S2 Movie, we found that the spatial and tem-poral patterns of wall shear rate ($\dot{\gamma}$), that is the ratio between the WSS and the viscosity, were very similar. Consequently, WSS results for the simulations with blood viscosity could be obtained simply by scaling up WSS results for the simulations with plasma viscosity by a con-stant factor that is the ratio of the two viscosity magnitudes. This is due to the very low Rey-nolds number nature of the flow in the embryonic heart, and the assumption that ventricular wall motions were driving the flow. We have previously verified this phenomenon in our study performing simulations on non-trabeculated hearts as well [17].

Results indicated that WSS were high at the regions near the inlet and outlet, due to high flow velocities and narrow channel dimensions at these locations. In addition, trabeculation surface structures caused high spatial variability of WSS, as previously reported [9,18], which was lower in the trabeculation grooves and higher at the ridges. This was more pronounced at the outer curvature of the heart where trabeculations were mostly found, and less at the inner curvature where trabeculations were absent. At the mid-diastolic time point, the region down-stream of the inlet on the inner curvature exhibited higher WSS than on the outer curvature. This corroborated past experimental findings that at low Reynolds numbers, high WSS were

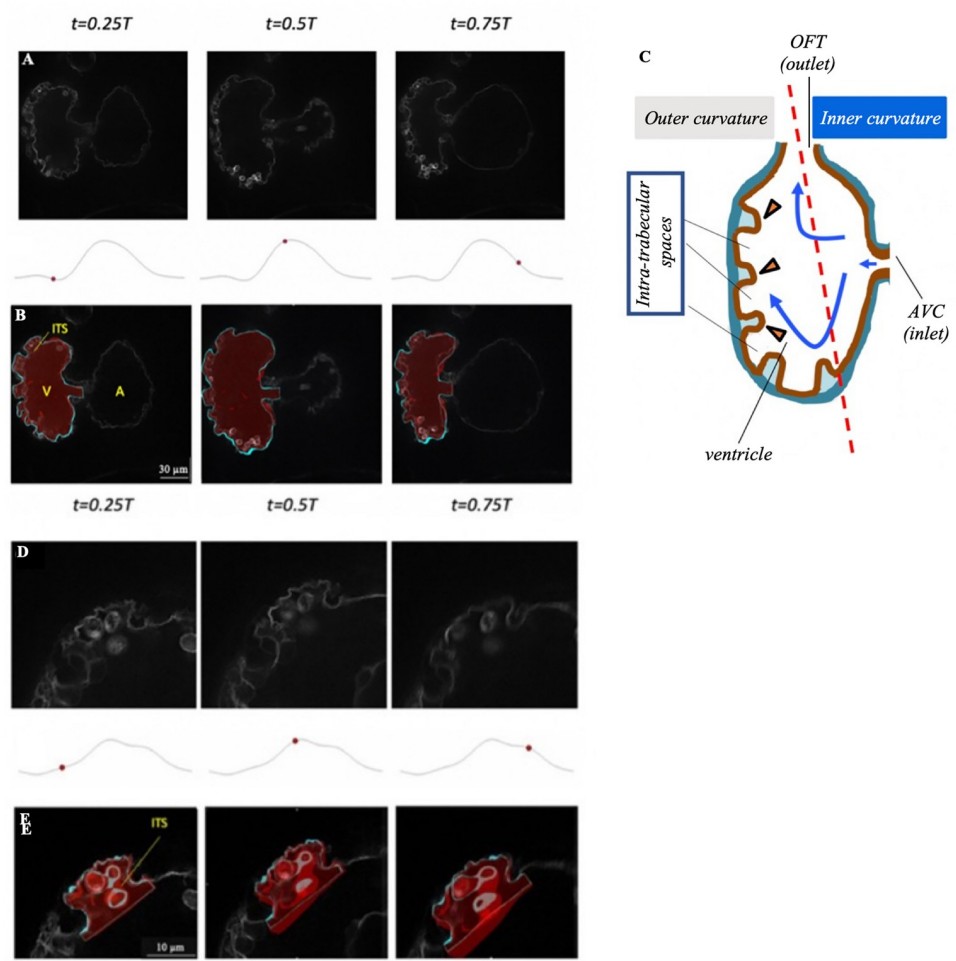

**Fig 1. Microscope images and anatomic 3D models of the developing zebrafish endocardium. (A,D)** Raw microscope images of (A) the whole ventricle, and (D) a single intra-trabecular space at 25%, 50% and 75% of the cardiac cycle. **(B, E)** Segmentation of (B) the ventricle and (E) a single intra-trabecular space, superimposed on the raw images, at the same time points as (A) and (D) (also shown in S1 Movie). The 3D reconstructed volumes are in red, while the regions of the 3D reconstruction close to the plane of the shown image are plotted in cyan on single 2D slice extracted from 4D image stacks of a zebrafish embryo from the *Tg(fli1a:Gal4ff;UAS:EGFP-CAAX)* line at 3 dpf. **(C)** Schematization of the intra-trabecular geometry, adapted from [15]. A, atrium; V, ventricle; ITS, intra-trabecular space. Here, t denotes time, while T denotes cardiac cycle duration.

typically found at the inner curvature of flow in curved channels [20], and findings in previous simulations of embryonic hearts [17,21]. The reason for this was that under the highly laminar and viscous flow environment, flow directions are easily bent by adverse pressure gradient in a curved channel, resulting in velocities being higher at the inner curvature than at the outer curvature and causing higher shear stresses at the inner curvature. If Reynolds numbers were higher, then fluid momentum will cause velocities and WSS to be higher at the outer curvature, but this was not the case here.

From our images of embryonic hearts in which both red blood cells and endocardial membranes were fluorescently labelled, we found that blood cells are not likely to enter the intra-trabecular spaces due to their small size (S3 Movie). As such, the trabeculation grooves were likely to experience plasma viscosity while surfaces elsewhere were likely to experience blood viscosity. In consideration of this, we proposed the mixed-viscosity scenario, where we

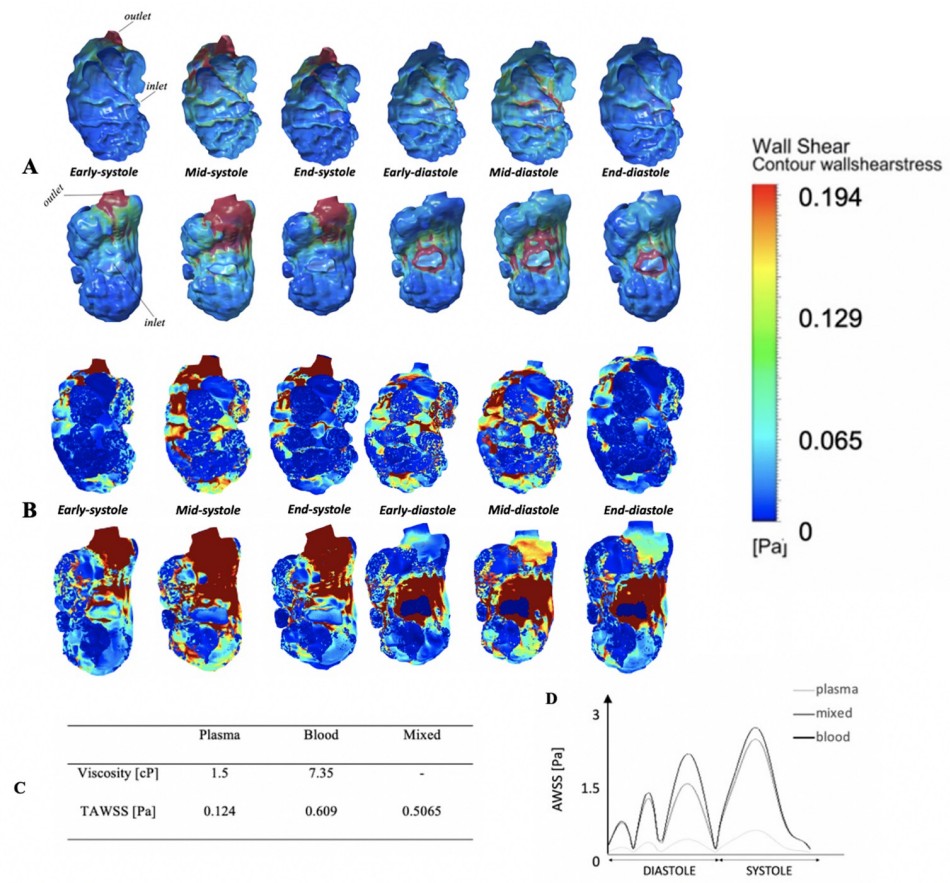

**Fig 2. Endocardial WSS profiles compared at different cardiac phases, with plasma viscosity and with the mixed-viscosity. (A)** Contour maps of endocardial WSS in a representative embryonic ventricle, over the cardiac cycle, with the assumption that fluid has the viscosity of plasma (1.5cP). Top row: ventral view of the outer curvature of the ventricle; bottom row: dorsal view of the inner curvature. **(B)** The same contour map of WSS, but for the mixed-viscosity assumption, where WSS results for plasma viscosity were adopted in the intra-trabecular spaces, but WSS results for whole blood viscosity were adopted for the trabeculation ridges. This scenario is generated considering that red blood cells do not enter the narrow intra-trabecular spaces often. **(C)** Time-averaged WSS over the ventricular surface calculated in simulations where the fluid viscosity was assumed to be that of plasma, blood, or mixed. **(D)** Temporal waveform of WSS averaged over the entire ventricular surface for the three scenarios.

obtained WSS results of the trabeculation grooves from the simulation conducted for plasma viscosity, and the WSS results elsewhere from the simulation conducted for blood viscosity, and combined them. This mixed-viscosity scenario was plotted as in Fig 2B and Fig F in S1 Text. The WSS magnitude differences between trabecular ridges and grooves were thus magnified, with the grooves experiencing much lower WSS than the trabecular ridges. We propose that this hybrid approach can be a good and novel approach to estimating WSS, short of performing complex biomechanical simulations of multiple individual blood cells within the fluid.

To characterize localized WSS behaviours, we compared WSS for 4 typical intra-trabecular grooves to that for 4 typical trabecular ridges immediately next to these grooves (Fig 3A and 3B). This was done via manual delineation of the obviously indented regions from the 3D reconstructed ventricle. Results showed that ridges experienced time-averaged WSS (TAWSS) that were two orders of magnitude higher than that in the grooves (Fig 3C). This can be broken down into wall shear rates being 3.1-fold higher, and viscosity being assumed to be 4.9-fold

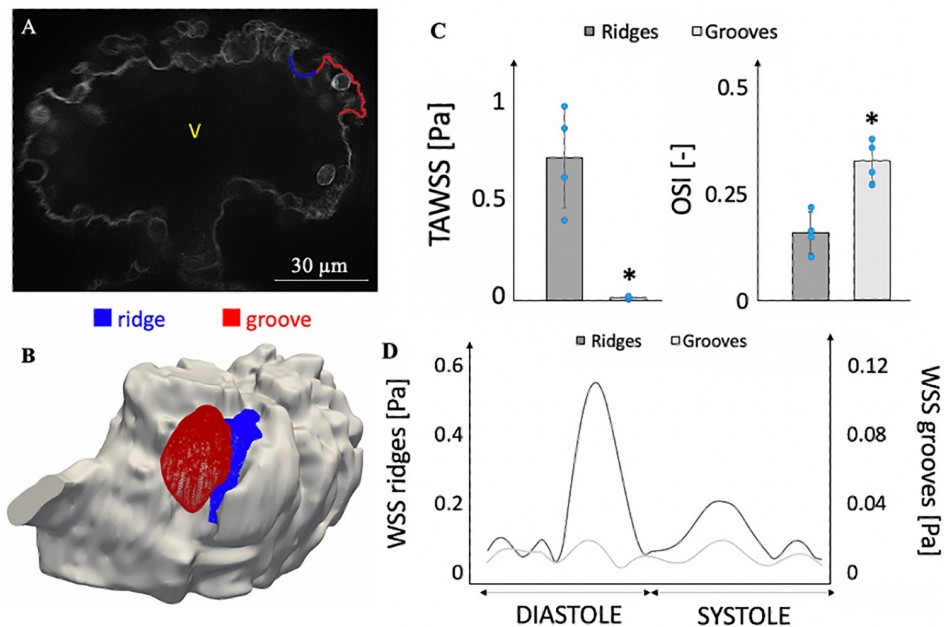

**Fig 3. Comparison of area-averaged WSS, time- and area-averaged WSS and OSI for the ridges and grooves. (A, B)** A typical intra-trabecular space (red) and the neighboring trabeculation ridge (blue) used for the WSS quantification and comparison. **(C)** Mean and standard deviation of the time-averaged WSS (TAWSS) and OSI (n = 4) with the mixed-viscosity assumption (viscosity of plasma for intra-trabecular spaces and the viscosity of blood for trabeculation ridges). Wilcoxon signed-rank test showed the lowest possible significance with the sample size (n = 4), p = 0.0625. **(D)** Temporal variation of WSS on a single ridge and groove under the same assumption.

higher. The intra-trabecular surfaces also experienced significantly higher oscillatory shear indices (OSI, defined in S1 Text), indicating higher oscillatory nature of their flow stresses, as previously reported [18].

The temporal waveforms of a typical intra-trabecular groove and trabecular ridge surface are shown in Fig 3D. At the trabeculation ridges, WSS typically showed two transient periods where WSS were elevated above the background of low magnitude and oscillating WSS waveforms. The first transient occurred during the late-diastolic A-wave filling phase, when atrial contraction sent high velocity flow into the ventricle. The second transient occurred during early systolic, when rapid contractions generated faster ejecting flow. This waveform corroborated earlier simulation studies [18]. The temporal WSS waveform of the intra-trabecular surfaces, however, was devoid of any transient periods where WSS was obviously elevated, maintaining the same pattern of low magnitude and oscillating WSS over the entire cardiac cycle.

## Effects of endocardial boundary motion dynamics on intra-trabecular WSS

Next, we investigated what factor or mechanism was important for bringing about fluid stresses on the endocardial surface of the intra-trabecular spaces. Fig 4 and Fig D in S1 Text showed results of our investigation.

Simulation results showed that detaching the main ventricular chamber from the intra-trabecular space to prevent shear interaction of fluid in these two chambers (changing from "Baseline" to "No Ventricle" scenario) caused changes to the spatial pattern of WSS, but no significant changes to the WSS magnitude averaged across all cases. Among the 6 intra-trabecular spaces investigated, spatially- and temporally-averaged WSS increased in some

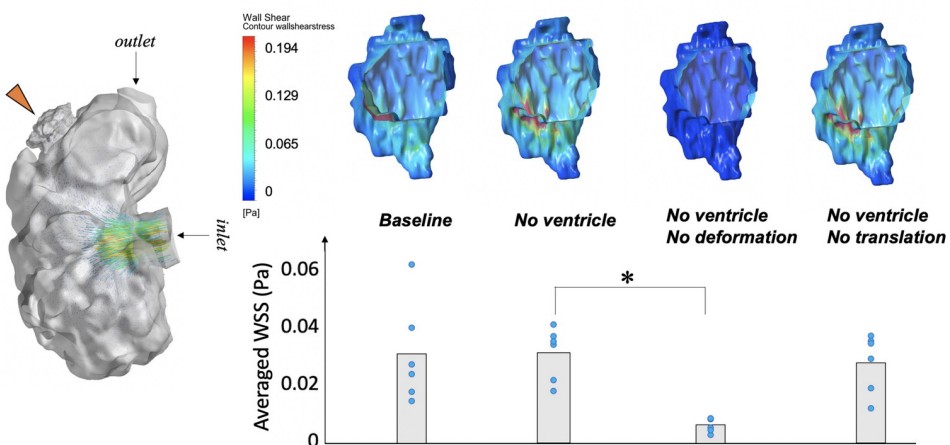

**Fig 4. Endocardial WSS for a single intra-trabecular space, or groove, under the various scenarios considered. (A)** Velocity vectors in a simulation conducted with the individual intra-trabecular space (orange arrow) joined to the main ventricular chamber for a 3 dpf zebrafish embryonic ventricle, which is the "Baseline" scenario in B. **(B)** Contour maps of the end-systolic WSS of the single intra-trabecular space under the different scenarios. In the "No Ventricle" scenario, the ventricular main chamber was detached from the intra-trabecular space, and replaced with a zero-reference pressure boundary condition. In the "No Ventricle, No Deformation" scenario, deformational motions, due to the contraction, were further removed from the "No Ventricle" scenario, but not the full translational motions. In the "No Ventricle, No Translation" scenario, translational motions were further removed from the "No Ventricle scenario", but not the deformational motions. **(C)** Mean and standard deviation of the WSS (n = 6) for each scenario, calculated by averaging both spatially (over entire endocardial surface) and temporally (over entire cardiac cycle). * $p < 0.05$.

cases and decreased in others. This suggested that the shear interaction between fluid in the intra-trabeculation spaces and flow in the main ventricular chamber was not necessary to generate intra-trabecular flow and WSS magnitudes, even though it affected WSS spatial-temporal patterns.

Next, when deformational motions caused by myocardial contractions were further removed from the individual intra-trabecular space after the detachment of the ventricular chamber (the "No Ventricle, No Deformation" scenario), the time- and spatial-averaged WSS were drastically diminished. However, when translational motions were removed after the detachment of the ventricular chamber (the "No Ventricle, No Translation" scenario), there were insignificant changes to WSS magnitudes and spatial patterns. This suggested that the squeezing motion of the endocardial boundary was the main driver of intra-trabecular flow and WSS, and the translation motion of the space had minimal contributions to flow and WSS. Here, rigid body rotations were not considered because such rotations were small (less than 0.07 radians with respect to end-diastole).

## Effect of cells in the intra-trabecular spaces

Interestingly, from our images and 3D segmentations, several cells with spherical shapes were found within the intra-trabecular spaces. These cells were located clearly apart from the endocardial boundary, and clear tissue connections to the endocardium was mostly not observed. However, their motion dynamics suggested that they were still attached to the endocardial surfaces, because they moved in synchrony with the wall without getting washed away, wobbling about their position within the intra-trabecular space (Fig 5 and S3 Movie). Moreover, there were no vortical flow patterns within the intra-trabecular space that could account for a convected cylic motion the cells if they were unattached. Further, the cells reverted to their initial locations with high precision at the end of a cardiac cycle, and displayed little randomness. As

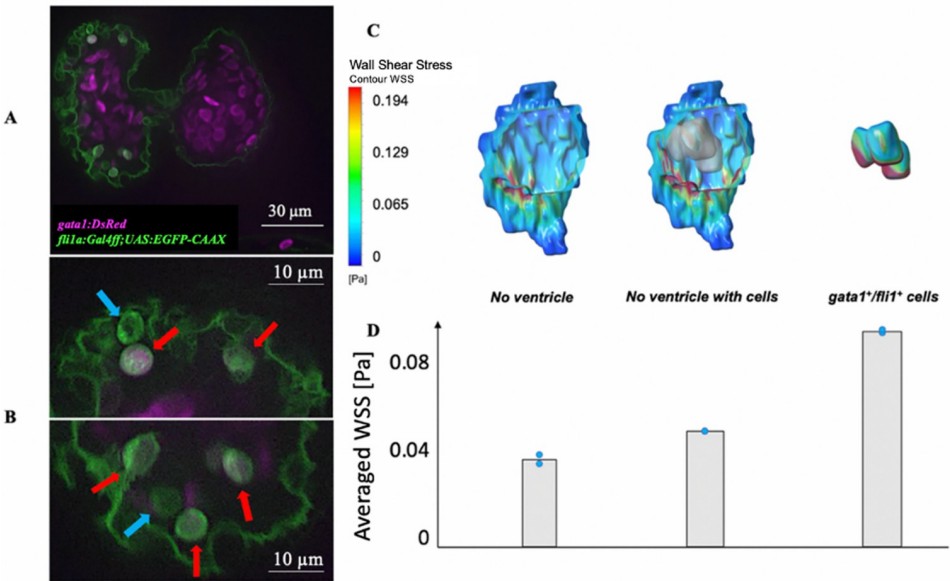

**Fig 5. Effect of trapped *fli1*+/*gata1*+ cells on the endocardial WSS. (A)** Confocal image of a 3 dpf zebrafish embryonic heart that was crossed between the *Tg(fli1a:Gal4ff;UAS:EGFP-CAAX)* line and the *Tg(gata1:DsRed)* line. **(B)** Close up of the image showing several wobbling cells appear to be trapped within intra-trabecular spaces from the image. Some of these cells were *fli1*+ (blue arrows), while some were *gata1*+/*fli1*+ (red arrows). These were thus hypothesized to be developing hematopoietic cells. **(C)** Flow simulation WSS results with and without the hematopoietic cells at end-systole, demonstrating that the cells elevated endocardial WSS. **(D)** Average WSS magnitude (n = 2) from the simulations, calculated by averaging both spatially (over entire endocardial surface) and temporally (over entire cardiac cycle).

such, we believed that it was unlikely that the cells were kept to their positions only by fluid lubrication forces, but nonetheless, we acknowledge that this remained a possibility.

These cells could exist individually, or cluster in groups of two to three. Interestingly, some of these cells were *fli1* positive, suggesting an endocardial lineage, while some were positive for both *fli1* and *gata1*, which might suggest a transition to become blood cells (Fig 5A and 5B). We thus hypothesize that these were endocardial hematopoietic cells, as reported previously in the literature [22], in various stages of transitioning from endocardial cells to blood cells. None of these attached cells were positive for *gata1* and negative for *fli1*. This could be because such cells would have fully detached and were washed away.

Since these cells occupied a significant percentage of the total volume of the intra-trabecular spaces, they were likely to influence the flow patterns and WSS within the spaces, and needed to be considered. We conducted simulations for intra-trabecular spaces disconnected to the main ventricular chamber, with and without these cells. Results show that the cells increased velocities in the intra-trabecular spaces and elevated the WSS on the endocardial surface (Fig 5C and 5D and Fig E in S1 Text). We had further conducted simulations of the groove space connected to the whole ventricle (the "Baseline" scenario), with and without the trapped cells and observed similar results whereby the presence of the trapped cells increased WSS (Fig H in S1 Text). This suggested that their presence enhanced the intra-trabecular squeeze-flow effects to hasten velocities and to lead to elevated WSS.

Further, previous studies have suggested that these *gata1*+/*fli1*+ cells could be mechanosensitive, and could rely on fluid forces stimuli to undergo the haematopoiesis process [22]. As such, we quantified the WSS on these cells as well, which were roughly double of the WSS on the intra-trabecular endocardial surfaces (Fig 5D).

## Effects of the trabeculation surface structure geometry on flow forces

To understand the role of trabeculation geometry on flow forces, we compared simulations results of the original trabeculated whole ventricles, and those with trabeculations removed and surfaces smoothed. The same motion field and the assumption of plasma viscosity was applied to both scenarios.

Results of non-trabeculated scenario are shown in Fig 6B (and Fig C in S1 Text), which can be compared to Fig 2A for the trabeculated scenario. At the inner curvature surface, WSS were largely similar in the two scenarios, due to the lack of trabeculations on this surface. On the outer curvature surface, WSS on the smoothed ventricle were similar to those on trabeculation ridges of the trabeculated ventricle, which were much higher than the WSS in the intra-trabecular surfaces. This is due to trabeculation ridges providing a sheltering effect for the intra-trabeculation fluid from the stronger flow velocities in the main ventricular chamber, and it corroborated previous findings [9,18]. The area- and time-averaged WSS on the trabeculated and smooth models were 0.124±0.085 Pa and 0.147±0.105 Pa respectively, with a 16% difference between the two models. The presence of corrugated trabeculation structures thus reduced WSS for much of the endocardial surfaces, and generated substantial spatial variability of WSS.

Since flow in the intra-trabecular spaces are likely to be recirculating, we calculated the OSI as well. Results are shown in Fig 6C (and Fig B in S1 Text) for both the trabeculated and smooth ventricles. Intra-trabecular surfaces had higher OSI and WSS here were thus more oscillatory than in the trabecular ridges, where WSS were more unidirectional, which corroborated previous reports [9,18]. The trabeculation surface morphology thus induced spatial variability and oscillatory nature of flow stresses across the outer curvature of the embryonic ventricle. We believe that the high OSI in the intra-trabecular spaces was attributed to the changing flow direction between filling and ejection, rather than vortical structures within the spaces, as we could not see any significant vortex formation in the trabecular grooves.

In both the trabeculated and smooth ventricles, however, the apical cavity of the ventricular chamber, opposite to the outflow tract, had high OSI, regardless of whether the surface was on the ridge or groove of the trabeculations (Fig 6C). This was likely a consequence of the location of this fluid volume, which was behind the ventricular inlet and far from the ventricular outlet. Flow within this volume would be pointed towards the apex during diastole, but would reverse and point towards the outlet during systole (Fig 6A), leading to oscillatory WSS. In contrast, in the region between the inlet and outlet, flow tend to point towards the outlet during both systole and diastole, which would result in lower oscillatory WSS.

## Effects of the trabeculation surface structure geometry on endocardial deformational burden

Results of comparing systolic-diastolic endocardial area strains of trabeculated versus smoothed ventricles are shown in Fig 6D (and Fig B in S1 Text), showing that the average surface strains were significantly reduced from 25% in non-trabeculated ventricles to 17% in the trabeculated ventricles. The stroke volumes of both ventricles were not significantly altered, and were in fact higher for the trabeculated ventricles. This suggested that to achieve the same contractile motion, the trabeculated heart did not need to expose endocardial cells to as much deformational stresses as the non-trabeculated heart. This thus supported our hypothesis that the trabeculation morphology enabled a reduction of endocardial strains during fluid pumping. We suggest that this is related to the physics of the corrugated surface having greater deformational flexibility than a smooth surface, as the corrugated surface can move without strains, such as straighten out or become more creased to change the ventricular shape, and did not need to rely on surface strains for this. Reduced deformational burden could pose an

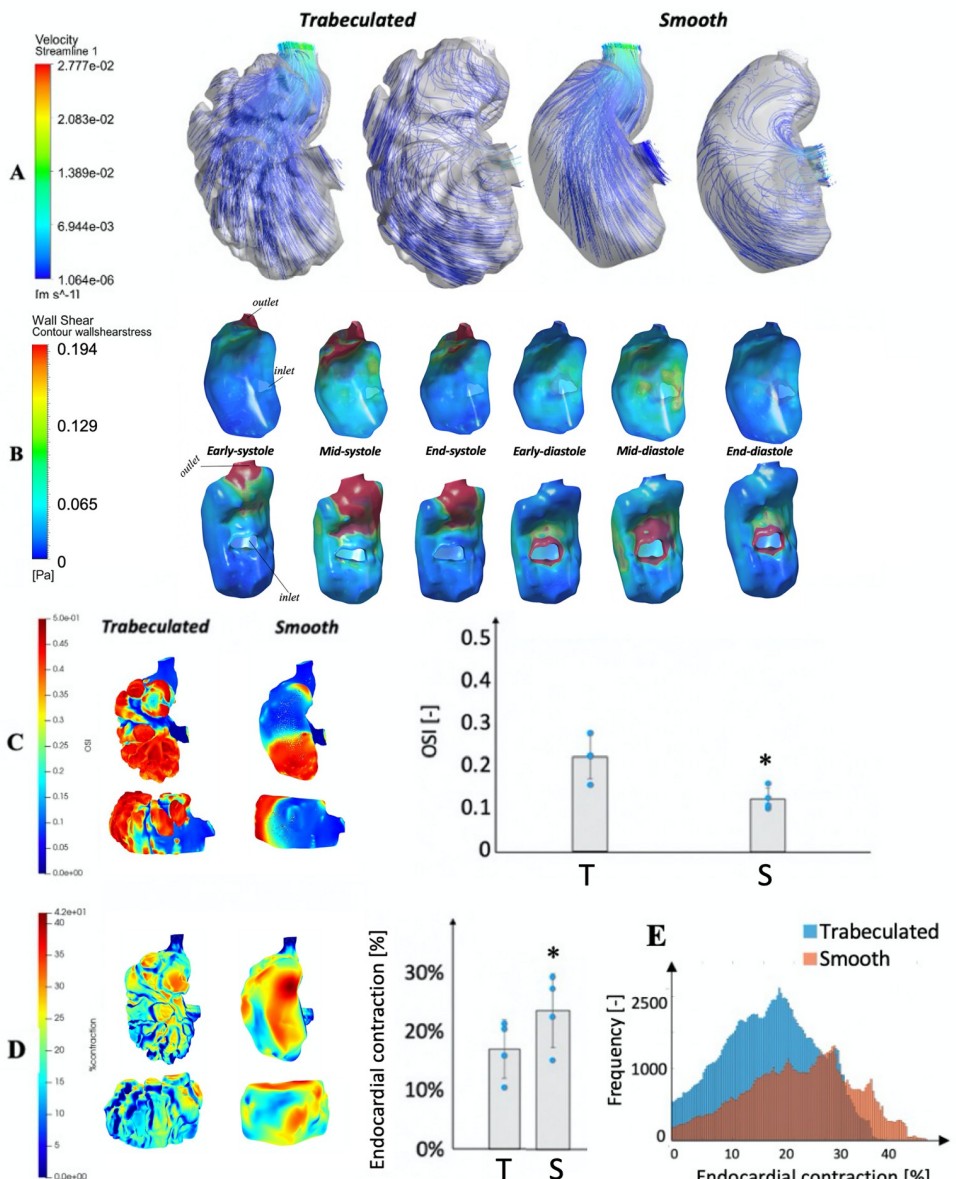

**Fig 6. Comparison of hemodynamic parameters between traeculated and smooth embryonic ventricles. (A)**
Velocity streamlines for both trabeculated and smoothed wall simulations of a 3dpf zebrafish embryonic heart from
lateral views, at (left) the end-diastolic phase and (right) the mid-systolic phase. **(B)** Contour maps of endocardial WSS
in the same embryonic ventricle of Fig 2, but with a totally smooth geometry, over the cardiac cycle with the
assumption that fluid has the viscosity of plasma (1.5cP). Top row: ventral view of the outer curvature of the ventricle;
bottom row: dorsal view of the inner curvature **(C)** Spatial pattern and surface-averaged magnitudes of oscillatory
shear index (OSI) for both trabeculated (T) and smoothed (S) wall simulations of a 3dpf zebrafish embryonic heart,
from lateral and ventral views. **(D)** Spatial pattern and surface averaged-magnitudes of endocardial contractile surface
area strains (end-diastole to end-systole), for both the trabeculated (T) and smoothed (S) simulations, from the same
views. **(E)** Histograms of the endocardial contraction across surface locations for the trabeculated and smooth models
of the same ventricular chamber.* p values were found to be at the minimum possible with the small sample size (*n* = 4,
*p* = 0.0625).

advantage to endocardium by reducing cyclic stretch injury. However, more comprehensive
evaluation of strains of the entire endocardial layer instead of just the endocardial membrane
surface is necessary to confirm this.

## Discussion

In the current study, we performed moving-wall CFD simulations based on high resolution spinning-disk confocal microscopy of a zebrafish embryonic line that labels endocardial cell membranes. This allowed us to clearly define the endocardial-blood boundary and to include fine anatomic and motion dynamic details of trabecular structures in our simulations, so as to obtain a more realistic estimation of the endocardial WSS in the zebrafish embryonic ventricle.

By using fish lines with fluorescence signals in the endocardial cell membranes, we found that our segmented ventricular model had visible differences from those in the literature where fish lines with fluorescence in the myocardium were used [9,15,18]. We also found that the calculated WSS, for the whole ventricle and single trabecular grooves and ridges, to be higher than those reported by these previous studies using myocardium fluorescence, even though we had used similar techniques as them. In the embryonic heart, the endocardial thickness was approximately 5 μm at each side of the lumen. The combined endocardial thickness at any cross section was thus approximately 10 μm, which was substantial compared to the ventricular inner diameter (30–50 μm). Omitting the endocardial layer could thus lead to substantial discrepancy in the WSS calculated at the trabecular ridges and non-trabeculated surfaces. In the intra-trabecular spaces, the endocardial layer thickness was of similar order of magnitude as the dimensions of these spaces, and the effects of omitting them would thus be similarly significant. In our results, we had also found that the squeezing motion of the intra-trabecular surface on the fluid within the space was very important for the generation of WSS on this surface. With the omission of the endocardial layer, it was likely that this squeeze flow could not be as accurately modelled. We thus believe that a more careful capture of the exact endocardial boundary location to be important for WSS quantifications. We further believe that it is important to conduct 3D, moving wall simulations, because the trabeculation geometry is very much 3D in nature, and our results indicated that consideration of the wall motion was important to WSS estimations.

In our study, we considered a mixed-viscosity scenario, where blood viscosity was assumed at the trabecular ridges and the plasma viscosity was assumed in the intra-trabecular surfaces. From our images, blood cells' diameters were about 6–8μm, which was comparable to the average width of intra-trabecular spaces, making it improbable that blood cells would enter the spaces. We did not observe any such events in our images, an example of which was given in S3 Movie. For this reason, it was reasonable to assume that the fluid in the intra-trabecular spaces to be purely plasma. However, due to the presence of blood cells in the main ventricular chamber, it would be reasonable to assume blood viscosity there. We thus believe that this mixed-viscosity scenario could be a good estimate of actual WSS environments in the embryonic heart with single-phase fluid simulations. However, we acknowledge that the best way to formally calculate WSS would be to conduct simulations that models individual blood cell and fully considers their biomechanics within the fluid.

Another important finding from our study was that the squeezing effect of the intra-trabecular endocardium on fluid within the trabecular groove space was the main driver of endocardial WSS within the space. The intra-trabecular fluid space should thus not be thought of as a passive space where flow and WSS is driven by the shear interaction of fluid with the main ventricular chamber. Rather, it is very active, and generates most of its flow and endocardial WSS. Since the prevailing belief is that appropriate flow stresses is necessary for proper development, we speculate that this intra-trabecular squeeze flow effect enables the ventricular contractions to exert the appropriate WSS on the endocardium for the correct development.

There is a lot of evidence in the literature supporting the idea that trabeculation depends on the presence of flow WSS stimuli. In embryos where heartbeat was pharmacologically stopped [13,23], and in mutants with weak flow in the ventricle [10,18], and hence low WSS, trabeculation was impeded. Consequently, it would be tempting to extrapolate these results, and use spatial differences in WSS environments to explain why trabeculations develop only at the outer curvature of the ventricle and not at the inner curvature. Our results, however, conflicted with this notion. If we used our smoothed ventricular model as a representation of the pre-trabeculation ventricle (Fig 6B), we could observe spatial patterns of WSS characteristics that did not coincide with sites of trabeculation. At the inner curvature, strong WSS was observed between the inlet and outlet, while weak WSS was observed between the apex and the inlet, but trabeculation did not develop on any of these surfaces. At the outer curvature, strong WSS was observed at the mid-ventricle and near outlet regions, but weak WSS was observed near to the apex, but trabeculations developed on all of these surfaces. The oscillatory nature of WSS did not coincide with the sites of trabeculation formations either. OSI was high at the apical region, at locations that eventually trabeculated as well as those that did not. These results together suggested that, although WSS stimuli was found to be necessary for trabeculation formation, it seemed that it was not the only determining factor. Other factors such as spatial variability of gene expressions, cell lineage, or perhaps mechanical deformation characteristics, could play a role in the initiation of trabeculation, as there is literature evidence of such variability. For example, Teranikar *et al.* found that the stretch deformations of myocardium were higher at the outer curvature than those at the inner curvature [24]. Further, Burkhard *et al.* performed spatially resolved RNA-sequencing of the zebrafish embryonic heart, and confirmed that substantial variability of gene expressions occurred along the length of the ventricle [25].

Investigators have recognized that the early embryonic circulatory system hosts sites of hematopoiesis, including the yolk sack [26], the dorsal aorta [27], and the heart tube [22]. At these sites, hematopoietic stem cells (HSCs) arise from specialized endothelial cells in a process termed the endothelial-to-hematopoietic transition (EHT) that involves activation of transcriptional programs necessary for HSC development [28]. These hemogenic endothelial cells have endothelial phenotypes and morphology, but also share many of the same cell surface markers expressed on HSCs [26]. Endocardial hemogenic processes had also been found to be essential for generating macrophages for proper heart valve formations [29]. Interestingly, the biomechanical forces were found to promote this emergence of HSCs in mouse embryos via mechanosensing mechanisms involving *Wnt*, *Notch*, and calcium fluxes [30].

In our study, we have identified cells within the intra-trabecular spaces that expressed both endothelial and blood cell markers, and thus we speculated that they are hemogenic cells. These cells that interacted with fluid within these spaces and exerted an influence on the endocardial WSS, which may suggest that proper quantifications of endocardial WSS require complex considerations of their presence. Since flow forces were shown to modulate embryonic endothelial hematopoeisis [30], it is possible that the fluid mechanics described in our study was important to the hematopoeisis process. Most of these cells were within the intra-trabecular spaces and at the apical region of the ventricle, where WSS were lower and more oscillatory, which might imply a specific requirement for flow mechanical environment for their formation. Further studies to test these notions seemed warranted.

In conclusion, our results suggest that a complex multi-component consideration, including fine-scale details of the endocardial cell membrane boundary location, squeeze-motion and cells attached to the endocardial surface, is required for the estimation of endocardial WSS. These complex interconnected features may be a mechanism for generating flow stimuli needed for development. Further, since mechanobiological processes are important for the embryonic heart development, our findings may be useful for future investigations on heart

development. Our results further suggested that, although it is well-known that flow stimuli are needed for trabeculation formation, additional factors are needed to explain why some parts of the ventricle trabeculate and others do not.

Our study has the following limitations. Firstly, our simulations assumed that blood was a continuum, but at this small scale, the embryonic blood was a two-phase fluid composed of blood cells and plasma. Our simulations were thus approximations and might have errors. With the two-phase fluid simulation model that fully considers blood cell biomechanics, we will likely observe that WSS will have more temporal unsteadiness, as WSS will be higher when a red blood cell was passing by, but lower when there was none, and thus a lager range of WSS could be observed. Secondly, we adopted a manual approach towards delineation of trabecular groove and ridge zones, which was arbitrary and might cause errors. Lastly, during contractile motion, some endocardial lining of the intra-trabecular spaces could undergo folding and contact, but this could not be fully captured by our motion tracking, and was modelled as a simple reduction in intra-trabecular surface area instead. This potentially created errors in WSS estimations.

## Materials and methods

### Ethics statement

Animal experiments were approved by the Animal Experimentation Committee of the Institutional Review Board of the Institut de Génétique et de Biologie Moléculaire et Cellulaire (IGBMC) in Strasbourg, France (reference number: MIN 4669–2016032411093030), following the European directive 2010/63/EU.

### Zebrafish line and imaging

The zebrafish lines used in this study were *Tg(fli1a:Gal4ff;UAS:EGFP-CAAX)* [31] crossed with *Tg(gata1:DsRed)* [32]. In the first line, membrane targeted EGFP is expressed in the endocardium, while in the second line, DsRed is expressed in blood cells. Fluorescent imaging of the beating heart was performed using a Leica DMi8 combined with a CSU-X1 (Yokogawa) spinning at 10 000 rpm, two simultaneous cameras (TuCam Flash4.0, Hamamatsu) and a water immersion objective (Leica 40X, N.A. 1.1). The entire hearts of the embryos were then imaged in XYTZ mode at 100 frames per second (~25 frames per cardiac cycle), with 10 milliseconds exposure time, and 2μm spacing between imaging planes. 4D images were reconstructed with voxel sizes at 0.186 x 0.186 x 2 μm$^3$ for the first embryo and 0.372 x 372 x 2 μm$^3$ for the other three embryo. Further details are given in S1 Text section 1.

### 4D reconstruction and motion tracking

Segmentation of the whole 3D ventricular chambers and intra-trabecular spaces were conducted with previous methods, [33] via a semi-automatic slice-by-slice approach using a custom-written lazy-snapping algorithm for pixel classification followed by Vascular Modelling ToolKit (VMTK, www.vmtk.org) for surface reconstruction. The models were trimmed and smoothed using Geomagic Wrap (Geomagic Inc., USA) for CFD simulations. Care was taken not to smooth out finer details of the ventricular endocardial boundaries. Next, cardiac motion tracking was performed using a well-validated cardiac motion estimation algorithm from our previous work, [19] which is briefly explained in S1 Text section 2. Segmentation was conducted only at one time point, and the reconstructed geometry could be animated to all other time points with the calculation motion field.

## CFD simulations setup

CFD geometry preparation, meshing and simulations were performed with ANSYS Workbench 19.4 (ANSYS Inc., USA). Meshing was done with density that exceeded the requirement indicated by our mesh convergence study. Details are given in S1 Text section 3. Dynamic-mesh CFD simulations were performed for 6 individual intra-trabecular spaces from 3 embryonic hearts, and for 4 whole ventricles. User-defined functions were employed to model wall motions according to our motion estimation algorithm.

Due to the consideration that blood cells do not enter intra-trabecular spaces often, simulations involving individual intra-trabecular spaces modelled fluid viscosity to be that of blood plasma. In the whole ventricle simulations, separate simulations for plasma and blood viscosities were conducted. Both fluids were assumed to be incompressible, Newtonian, and to have a density of 1.025 kg/L, but to have differing viscosity (1.5 cP and 7.35 cP [34] for plasma and blood, respectively).

In the whole ventricle simulations, the ventricular inlet was specified to be the zero-reference pressure boundary while the outlet was specified to be an impervious wall during diastole, and vice versa during systole. Flow in the ventricle was assumed to be driven by programmed wall motions as extracted from the images. Simulations were conducted on the original trabeculated ventricle reconstruction, and on the manually smoothed and non-trabeculated version, to understand the effects of trabeculation on flow dynamics.

For each single intra-trabecular space simulation, we manually detached the intra-trabecular space from the ventricle by cropping the ventricular chamber out along a single plane close to the inlet of the intra-trabecular space. The opening of the trabecular space left behind by the cropping was then specified as reference zero pressure boundary condition. Simulations were then conducted just for the single trabeculation space, independent of the main ventricular space. For these individual intra-trabecular space simulations, we investigated various scenarios, to understand what was driving flow and WSS in the inter-trabecular space. The "Baseline" scenario was when the intra-trabecular space was part of the ventricle, the "No Ventricle" scenario was when the space was when the ventricle was removed and replaced by zero-reference pressure boundary condition. We decomposed wall motions into translational and deformational motions, and further tested scenarios when either was removed. Further details are in S1 Text section 5.

## Endocardial deformational burden

We hypothesize that the corrugated surface geometry of the trabeculated heart allows the heart to perform its function with a reduced endocardial deformational burden. To test this, we compared the systole-to-diastole area deformation strains at the endocardial surface of the trabeculated whole ventricle models to that on the manually smoothed non-trabeculated version, using the same motion field obtained from images. Strain was obtained by calculating the percentage area change to each *.stl* surface triangular mesh element from end-diastole to end-systole. Trabeculation removal was performed by applying aggressive smoothing in Geomagic Wrap®. In other words, we are investigating how much endocardial contractile strain will be needed in the non-trabeculated heart to achieve fliud pumping motion and function as the trabeculated heart.

## Statistical analysis

All the results were expressed as mean ± SD. The Wilcoxon signed-rank test was used for hypothesis testing with $p<0.05$ considered significant.

## Supporting information

**S1 Text. Supplementary Text for Fluid Mechanics of the Zebrafish Embryonic Heart Trabeculation.**
(DOCX)

**S1 Movie.** Motion tracking of the 3D reconstruction of the (left) single intra-trabecular space and (right) whole ventricle of the zebrafish embryonic heart in red, superimposed onto a single plane of the 3D microscopy image, to show a satisfactory segmentation and motion tracking. Green zones are portions of the 3D reconstruction that are close to the image plane displayed.
(MP4)

**S2 Movie. Results of the whole ventricular simulations, showing WSS surface countour plots for both the plasma viscosity and blood viscosity scenarios.**
(MP4)

**S3 Movie. Spinning disc confocal microscopy images of zebrafish embryonic heart with gata1 and fli1a fluorescence labels, demonstrating a populations of cells that are positive for both markers in the heart, which are likely hemogenic cells.**
(MP4)

**S1 File. User defined function file used in the ANSYS CFD Simulations.**
(C)

## Author Contributions

**Conceptualization:** Adriana Gaia Cairelli, Julien Vermot, Choon Hwai Yap.

**Formal analysis:** Adriana Gaia Cairelli, Renee Wei-Yan Chow, Choon Hwai Yap.

**Funding acquisition:** Julien Vermot, Choon Hwai Yap.

**Investigation:** Renee Wei-Yan Chow.

**Methodology:** Adriana Gaia Cairelli, Choon Hwai Yap.

**Project administration:** Choon Hwai Yap.

**Supervision:** Julien Vermot, Choon Hwai Yap.

**Writing – original draft:** Adriana Gaia Cairelli, Choon Hwai Yap.

**Writing – review & editing:** Adriana Gaia Cairelli, Renee Wei-Yan Chow, Julien Vermot, Choon Hwai Yap.

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
