## [Decision Letter · Decision Letter 0]

17 Mar 2022

Dear Dr. Yap,

Thank you very much for submitting your manuscript "Fluid Mechanics of the Zebrafish Embryonic Heart Trabeculation" for consideration at PLOS Computational Biology.

As with all papers reviewed by the journal, your manuscript was reviewed by members of the editorial board and by several independent reviewers. In light of the reviews (below this email), we would like to invite the resubmission of a significantly-revised version that takes into account the reviewers' comments.

We cannot make any decision about publication until we have seen the revised manuscript and your response to the reviewers' comments. Your revised manuscript is also likely to be sent to reviewers for further evaluation.

Sincerely,

Alison L. Marsden

Associate Editor

PLOS Computational Biology

Daniel Beard

Deputy Editor

PLOS Computational Biology

Reviewer's Responses to Questions

**Comments to the Authors:**

Reviewer #1: The authors simulated the blood flow of 3dpf embryonic zebrafish heart to elucidate the effect of ventricular geometry. Their results were improved from previous research by including endocardial layers which is direct contact boundary of blood. Although WSS increased after including endocardial layers (~10um thick), interestingly, WSS pattern remain similar which could validate their simulation results. In addition, they tried to understand what causes the driving flow and WSS of the inter-trabecular spaces. They simulated the different scenarios of geometry of inter-trabeculae and found that the inter-trabecular squeeze flow effect is the main driver and translational motion of space doesn’t significantly contribute to the flow and WSS. Another interestingly finding is that their endothelial-to-hematopoietic transition could happened in inter-trabeculae regions. Based on their spinning disk confocal, they observed cells that positively expressed both fli1 and gata1. Overall, this research could provide an important direction of the mechanotransduction in cardiac development using zebrafish. However, there are some concern should be address.

1. Authors measured the dimensions of zebrafish trabeculae as longitudinal, radial and circumferential dimensions were 12.5±3.1 μm, 123 8.7±0.8 μm, 21.7±10.7 μm (n=16) respectively. However, it is unclear where the author measured. Trabeculae mostly develop in outer curvature, but some are near apex and outflow tract. Trabeculae near outflow tract and apex should be relatively small. It would be helpful to address specifically where the author measured. Also, the measurement are in relaxation stage or contraction stage? Also, if you can make a table, it would be helpful to present.

2. Authors mentioned in line 177-180 that high WSS found at inner curvature compared to outer curvature due to low Reynolds numbers. Although he properly cited, it would be helpful to briefly explain what makes this event.

3. Authors should describe better how to simulate hybrid approach to calculate WSS of inter-trabeculae where blood cells can’t occupy while heartbeat. It is interesting approach without using two phase flow simulation. However, detail explain should be address in methods section

4. Direction of zebrafish heart in Fig. 2 and supplementary figures are not well depicted. For example, for Fig. 2A, outlets of the upper figures are pointing left side, but lower figures are opposite. Zebrafish heart directions should be consistent and clear. For supplementary figures, author didn’t point out where the outflow tract and inlet flow are. Thus, it took me a time to figure out where the area between inlet and apex.

Reviewer #2: In this study, the authors use computational fluid dynamics to analyze the fluid mechanics of trabeculations in developing zebrafish. Using a fish line that expresses endocardial lining, the study finds that trabeculations lead to a spatially varying and oscillatory shear profile. This is further attributed to a local squeezing flow that has minimal interaction with the bulk flow in the ventricular cavity.

The analysis is interesting and the manuscript is generally well written with some minor typos and sentence construction errors that could be rectified during revision. However, the quality of the manuscript could be substantially improved by furnishing additional details and clarifying some areas to avoid confusing the reader and facilitate reproducibility.

Major:

- One aspect that is ignored in this study is that trabeculations form a network. During deformation, the network could undergo `squeezing` motion as rightly predicted by the authors and also hypothesized by Ares Pasipoularides (Heart’s Vortex). This complex motion could lead to contacting surfaces and may provide additional contribution to stroke volume. In this study, however, the endocardial lining is extracted as a continuous surface that remains intact during the deformation. The complex squeezing motion of the trabeculations with contact is nearly impossible to model but should be addressed as a limitation of the approach.

- Page 4, Lines 92-95; Page 5, Line 110: It is not clear what the authors mean by optimized estimation of endocardial WSS. This needs to be clarified or rewritten.

Perhaps the focus could be on the mechanism for generating fluid forces in the intra- and inter-trabecular spaces?

- WSS characterization (Page 7, Lines 150-155): It is noted that the WSS for simulations with different viscosities can be obtained by simply scaling with the corresponding ratio, and this is attributed to the low Re. However, this could also be attributed to the flow being assumed to be Newtonian and therefore, viscosity is independent of shear rate.

While this is interesting and could avoid the need to perform complex simulations involving blood rheology, can the authors provide the range of shear rates observed in the ventricle? Does assuming non-Newtonian flow change the viscosity substantially at low shear rates? If so, then simple scaling of WSS due to change in viscosity may not be possible.

- Inter-trabecular analysis (Page 11, Lines 225-232): The setup of analyzing individual trabeculations is intriguing, especially the zero-pressure boundary condition. This boundary condition would not capture shearing motion between the bulk ventricular flow and the flow in the trabecular spaces.

Further, when the trabeculations expand and contract, they would exchange flow with the ventricular cavity. Therefore, it is misleading to suggest that the interaction is not necessary to generate inter-trabecular flow (lines 230-232).

More details need to be furnished on how the isolated trabeculations simulations are set up. How is the ventricle detached or attached to the trabeculations? Where are the boundary conditions applied? Is there no flow at all in the bulk cavity in the no-ventricle case? Are the individual trabeculations solved?

For the sake of completeness, can the authors perform analysis on OSI for the individual trabeculations?

- On the effect of cells in the inter-trabecular spaces, it is not clear how these cells are included/excluded in the analysis. Are they introduced in the model as finite obstructions that are connected to the outer wall? Are they individually tracked?

- On page 17, line 335, it is conjectured that recirculatory flow could be observed in the inter-trabecular spaces. As the authors here perform flow analysis in the individual trabeculations, can the authors use data to support this argument? IS the flow recirculatory in these cavities?

Is the higher OSI in these cavities attributed to the recirculatory flow or the changing flow direction between filling and ejection?

- In the Methods section, can more details be added on how the images scanned were synchronized to the cardiac cycle? Is the imaging volumetric or was scanning performed on a single plane for a certain time before advancing to the next plane?

- On the Data Availability, while the main CFD solver commercial and could not be shared publicly. However, the authors could provide access to images, ventricular models, any codes used to process the images and extract motion, any post-processing scripts, etc. Otherwise, this would seriously affect reproducibility of the results. Any restrictions on the data/code availability should be clearly specified in the manuscript.

Minor:

- Please thoroughly revise the manuscript to fix some minor spelling mistakes and sentence constructions errors.

- Abstract: something is missing in the sentence, "By comparing our results to literature..." please rectify.

- Can the authors cite refs 9 and 18 wherever spatial variability in WSS is discussed to support their arguments? Also, some references have co-first authors. Please rectify.

- The authors generally refer to `inter-trabecular' spaces in the manuscript for identifying the regions within the trabecular cavities. However, it might be better to use the word `intra-trabecular' to refer to these cavities and reserve `inter-trabecular' to the endocardial segment between the trabeculations.

- Fig 6C, 6D. please provide x-axis labels/legends for the bar charts.

- Page 20, lines 397-399, it is mentioned that blood viscosity is locally varied between trabecular ridges and in the inter-trabecular spaces. Can the authors confirm if the viscosity is spatially varied in the problem setup? Can this be highlighted in the Methods section?

Reviewer #3: This paper demonstrates a CFD-based shear stress analysis in embryonic zebrafish heart. Compared to most of previous studies, it adopts high resolution spinning-disk confocal microscopy and fish lines that label the endocardial cell membranes rather than myocardial cells. The ventricular domain, as well as the trabeculation ridges and grooves could therefore be defined in a more accurate fashion. The novel aspects are using an endocardial marker to better resolve trabecular structures, and the interesting result that squeezing motion is the main driver of flow in inter-trabecular spaces. Overall, the paper is interesting, although the methods do not appear to be particularly novel, and the analysis is fairly simple on the fluid mechanics side.

Major Comments

1. The use of the “mixed viscosity” model needs better justification. In the Stokes flow limit, structures in a flow have a long-ranged effect. Even though the red blood cells are not in the inter-trabecular spaces, they may still have a strong effect on the fluid mechanics in these spaces. In addition, the bulk viscosity assumption for blood is only valid when the length scale of the flow is much greater than the length scale of the blood cells. From the authors’ movies, the size of the blood cells is fairly large compared to the ventricle itself and is certainly on a similar scale to the inter-trabecular spaces. Uncertain is the bulk viscosity assumption, and further clarification is needed in the discussion.

a. The authors may consider performing a fluid-structure interaction simulation with fully-resolved RBCs to support the result with that of bulk viscosity assumption.

2. Line 351: This section about endocardial deformation is interesting, and further development or removal from the discussion as the manuscript seems to be primarily focusing on fluid mechanics.

a. The authors note that the surface strains were smaller in the trabeculated heart than the non-trabeculated heart. Isn’t this due to the higher surface area in the trabeculated than non-trabeculated heart?

b. Is there any spatial variability in surface strains? Higher/lower in the trabecular ridges/grooves?

Minor Comments

1. The authors may edit and address multiple typos.

2. Line 121: Please provide a schematic to explain the relevant geometry to the “inter-trabecular space”.

3. Figures 1, 2: Images are too small. It is difficult to see the important features. Seeing your supplementary movies, the full-size renderings are quite striking to see, so I would like to see them in the paper.

4. Line 143: In this section, define WWS and wall shear rate.

5. Line 222: In this section, please explain in detail the different scenario and how they are defined in the methods.

6. Line 251: In the case where translational motions were removed, were rigid body rotations also removed? Typically, motions are decomposed into translation, rotation, and deformation.

7. Line 261: The cells may be “attached” to the endocardial surface via lubrication forces.

8. Line 476: Please further discuss the limitations; that is, estimate how your approximations and assumptions affect the validity of your answer.

9. Line 507: Have you validated the motion tracking algorithm with this data by comparing the reconstructed geometry to manual segmentations at later time points in the cardiac cycle?

**Have the authors made all data and (if applicable) computational code underlying the findings in their manuscript fully available?**

Reviewer #1: Yes

Reviewer #2: **No: **The CFD solver employed in the study is commercial and may not publicly shared. However, even the images, computational models, codes for image processing and motion extraction, post-processing, etc. could be shared but are not available. Restrictions need to be specified in the manuscript.

Reviewer #3: Yes

PLOS authors have the option to publish the peer review history of their article (what does this mean?). If published, this will include your full peer review and any attached files.

Reviewer #1: No

Reviewer #2: No

Reviewer #3: **Yes: **Tzung Hsiai
---

## [Decision Letter · Decision Letter 1]

26 Apr 2022

Dear Dr. Yap,

We are pleased to inform you that your manuscript 'Fluid Mechanics of the Zebrafish Embryonic Heart Trabeculation' has been provisionally accepted for publication in PLOS Computational Biology.

Best regards,

Alison L. Marsden

Associate Editor

PLOS Computational Biology

Daniel Beard

Deputy Editor

PLOS Computational Biology

Please address the reviewers comments about editing for grammar and submit a final version.

Reviewer's Responses to Questions

**Comments to the Authors:**

Reviewer #1: Authors properly address all of my concerns

Reviewer #2: The authors have done a commendable job in addressing all the technical aspects of my previous review. However, grammar could be substantially improved before publishing in PLOS Comp Bio journal. For instance, there is more usage of `which’ in the manuscript, and the usage of articles could be improved. E.g. in the abstract, “...(WSS) results were found to exceeded those reported in existing literature...” should be “exceed”; a `the’ is missing in “...rather than the shear interaction with flow in the main ventricular chamber...”.

Reviewer #3: The authors adequately responded to the reviewer's comments.

**Have the authors made all data and (if applicable) computational code underlying the findings in their manuscript fully available?**

Reviewer #1: Yes

Reviewer #2: Yes

Reviewer #3: Yes

PLOS authors have the option to publish the peer review history of their article (what does this mean?). If published, this will include your full peer review and any attached files.

Reviewer #1: No

Reviewer #2: No

Reviewer #3: No

---

## [Editor Report · Acceptance letter]

27 May 2022

PCOMPBIOL-D-22-00195R1 

Fluid Mechanics of the Zebrafish Embryonic Heart Trabeculation

Dear Dr Yap,

I am pleased to inform you that your manuscript has been formally accepted for publication in PLOS Computational Biology. Your manuscript is now with our production department and you will be notified of the publication date in due course.

With kind regards,

Livia Horvath
